# The limits of nudging: Results of a randomized trial of text messages to promote home-based caregiving and reduce perceptions of stigma for COVID-19 patients in Kenyan informal settlements

**James B. Tidwell[1,2], Jessie Pinchoff[3]\*, Timothy Abuya[4], Eva Muluve[4], Daniel Mwanga[4], Faith Mbushi[4], Karen Austrian[4]**

**1** World Vision Inc., Washington, DC, United States of America, **2** University of North Carolina Gillings School of Public Health, Chapel Hill, NC, United States of America, **3** Population Council, United States–Social and Behavioral Sciences Research, New York, NY, United States of America, **4** Population Council, Kenya–International Program, Nairobi, Kenya

\* jpinchoff@gmail.com

**Data Availability Statement:** The questionnaires and datasets are available through Harvard's

## Abstract

During the early stages of the global COVID-19 pandemic, governments searched for effective means to rapidly disseminate information about how to prevent the disease and care for sick household members. In June 2020, the government of Kenya considered sending text messages, a behavioral nudging approach, to inform and persuade the public to practice home-based care for those who were infected. We conducted a randomized evaluation of simple informational messages compared to messages targeting personal and social benefits for those receiving the messages. We hypothesized that those that received messages tailored around social or personal benefit would be more likely to undertake the promoted behaviors of isolating if infected with COVID-19 and intending to care for an infected family member. While fear and perceptions of stigma were widespread, more than two-thirds of respondents in the control condition expressed an intention to care for an infected family member at home. Despite greater recall of the personal benefit message, which used reciprocity as its key behavioral lever, intentions to provide care at home and perceptions of stigma did not differ across study groups. Rather, capabilities such as wealth and having sufficient room at home were the key determinants. While text messages as behavioral nudges may be useful for some behaviors, policymakers should consider a broader range of tools for behaviors that are influenced by people's capabilities, since even low-cost interventions may crowd out the time and energy needed for other responses during an emergency.

## Background

COVID-19, a highly infectious disease that rapidly spread around the world, was first identified in Kenya in mid-March 2020, with cases rising from 81 on April 1ˢᵗ to more than 110,000

Dataverse. https://dataverse.harvard.edu/dataset.
xhtml?persistentId=doi:10.7910/DVN/VO7SUO.

**Funding:** This study received a small grant from
IPA's Peace and Recovery Program, through the
Department for International Development, UK
Government, MIT0019-X15 awarded to PI Karen
Austrian. The funders had no role in study design,
data collection and analysis, decision to publish, or
preparation of the manuscript. No authors received
a salary from the funders.

**Competing interests:** The authors have declared
that no competing interests exist.

by March 15[th], 2021 [1]. The Government of Kenya's initial response included a ban on international flights, closing schools, and banning large social gatherings to prevent super-spreading events. The Kenyan Ministry of Health launched a COVID-19 taskforce to steer the country's prevention, containment and mitigation measures [2]. As part of their response, the Kenyan government began to send out informational messages by various media, including television, radio, and text messages. These messages included information on the symptoms of COVID-19 and what to do if one experienced those symptoms. Initially, all positive cases were hospitalized and those who had travelled or had a known exposure were kept in isolation centers. As cases began to rise, the Kenyan Ministry of Health transitioned to recommending self-isolation at home in early June 2020. This had significant implications for household members that would be responsible for providing such care.

Infectious diseases present challenging decisions to households, families, and communities —failure to perform individually costly behaviors (such as staying home or caring for someone infected) may result in harm to larger groups (greater spread through the population or higher risk of mortality). In the context of the Ebola virus disease outbreak in West Africa starting in late 2013, both individual- and community-level factors related to fear of contracting the disease and perceived risk of future outbreaks were found to drive this stigmatization of survivors and their exclusion from social interactions [3]. The effects of the separation and isolation of suspected cases and fears of persistent infection risk took a significant toll on the mental health of individuals and communities and persisted for years afterwards [4]. It has been widely reported that initial COVID-19 messaging when the public had low understanding of the disease increased both the social isolation of infected patients and stigmatization of frontline healthcare [5–8].

We report findings from a study of adults in informal settlements in Nairobi, Kenya, including attitudes and practices related to caring for those infected with COVID-19 during the first months of the pandemic. We employed a behavioral nudging approach, sending text messages aimed at shifting behavior. Participants were randomly assigned to receive one of three versions of a text message meant to reduce social stigma around caregiving. The Government of Kenya broadcast informational text messages on a variety of aspects of COVID-19 during the first several months of the outbreak, as it was a rapid, low-cost approach to disseminating information. Our study used the same text message approach to ensure the opportunity for scale up. Text messages have been shown to impact a wide variety of public health behaviors [9]. While one of the most natural and effective uses of text messages is to provide timely reminders [10], they have also been shown to be effective when targeting behavioral motives [11]. If able to communicate normative expectations and that they've reached a large proportion of the population, they may plausibly be able to influence social norms as well [12]. However, some behaviors may require more than increased motivation, especially where the capability to do the behavior is significantly influenced by the setting [13]. More than 90% of respondents reported receiving text messages about COVID-19 in earlier rounds of data collection in the study cohort; but, information was being disseminated through an overwhelming number of modalities, as more than half also reported receiving information through government TV and radio ads; TV and radio programs; friends, acquaintances, and neighbors; and social media and the internet [14].

Given the community impact of individual behaviors to promote or prevent the spread of COVID-19, it is plausible that both personal motivations and prosocial motivations could influence relevant behaviors. A study in the United States found that while prosocial messages were effective in some stages of the pandemic, perceived societal threat was more strongly associated with intentions to perform preventive behaviors than perceived personal threat was [15]. A study in Denmark found messages promoting social distancing were more effective if

they used personal benefit motivations compared to social benefit motivations [16]. However, the simple dichotomy of prosocial vs. personal motivations obscures the large variation within each of these categories—for example, a broad range of personal and social motivations may function at different cognitive levels, varying from personal motives like hunger or curiosity to social motivations like affiliation, status, or justice [17]. A meta-analysis found that tailored messages with some personalization were significantly associated with greater intervention efficacy [8] though overall the effects of text messaging for health promotion interventions are mixed. At the time of the study, the guidance was to quarantine for 10 days, which required family members to care for an infected individual. Messaging was used to inform the public that they should care for family members and reduce stigma towards those infected.

As the government was already using basic text messages and was interested in the effectiveness of these to promote home-based care giving, the study team proposed to test some different variants to understand the scope of their potential impact. Based on existing literature, behavioral theory, and local contextual assessment by the study team, we tested the impact of a personal-benefit message (targeting reciprocity) and a social-benefit message (targeting affiliation with one's community and country) compared to a control message only providing information, to understand how we might improve caregiving knowledge and behaviors related to COVID-19 infection [17–20]. Our primary research question was: Can we improve knowledge, attitudes (stigma, specifically), and behavioral intentions related to the isolation of cases/caregiving for those infected with COVID-19 using a brief text-message-based intervention promoting individual or social motives? We hypothesized that text messages tailored with either social or personal benefit framing would improve the recall, attitudes, intentions, and perceptions of stigma regarding personal isolation if infected and of caring for a sick family member at home, in adherence with local guidelines at the time.

## Methods

### Behavioral intervention

In consultation with the Government of Kenya, we aimed to develop and test a few variations of an SMS message to see if recipients would report differences in how the information was perceived and recalled. We assessed the impact of two different behaviorally informed messages compared to a control message that simply stated the desired behavior. In the *control* condition, we sent recipients an information only message to share that all were susceptible infection: "Anyone can get infected with the Coronavirus (COVID-19). Those infected should stay isolated until recovered, but should still be loved, cared for, and accepted by friends and neighbors." This was relatively new information to the target population, given the government's policy of sending those infected to isolation centers up to that point. While knowledge may not be sufficient to drive behavior change, it is likely a pre-requisite in many cases, especially where motivational messaging is used.

In the *personal benefit* condition, we added to the beginning of the control message the sentence: "Treat others with Coronavirus how you would like to be treated." As we were examining a behavior that was unlikely to have direct personal benefit (isolation and caregiving being both individually costly behaviors), this condition focused on the idea of reciprocity. Reciprocity is ubiquitous in human cultures [18], has been shown to be an optimal strategy for cooperation in some settings [19], and may be one of the primary bases of effective public policy [20].

In the *social benefit* condition, we added to the beginning of the control message the sentence: "Supporting one another will help our community and nation through this difficult time." This approach draws upon the motivation of being included in one's community, sometimes called affiliation, which is a strong driver of norm compliance [17]. This was selected

because these behaviors, especially if infection presents with relatively mild symptom, may not be subject to direct normative pressures if others did not know that the person was infected.

While a range of personal and social benefits and a multitude of ways of phrasing each of these are possible, these messages were developed based on the theoretical levers thought most likely to impact the relevant behaviors and based on iterative feedback from local research staff and the Kenyan government. These messages are framed to increase recall and improve attitudes and uptake of the promoted behaviors.

## Study setting and sample size

This study was conducted as a part of a panel survey with a cohort constructed from two ongoing studies in five urban informal settlements in Nairobi, Kenya. Participants in the prior studies, Adolescent Girls Initiative-Kenya (AGI-K) (n = 2,565) and the Listen to Me, Let's Grow Together (called NISITU) (n = 4,519) were combined and a random sample drawn until about 550 households were selected from each informal settlement. More detail on sampling is available elsewhere, but in brief these participants were sampled from each informal settlement from households with adolescents, and had provided contact information for future studies [14]. Each individual gave verbal consent over the phone to participate. By round 4 of the survey, when the data that were the primary focus of this study were collected, the cohort still included 1,525 of the original approximately 2,000 people included. If a person did not respond to one round of the survey, attempts were still made to contact them for subsequent rounds, so that no one was formally dropped from the cohort unless they asked to be removed. To reduce the burden on participants of repeated surveys, a subset of 1,160 received the follow up survey with questions as part of the randomized controlled trial of the SMS messages. The final sample size was sufficient to compare each behaviorally informed study arm to the control with the power to detect a 10-percentage-point difference when controlling for multiple comparisons using a Bonferroni-Holm procedure [21].

## Design and procedures

All who were still included in the cohort were randomly allocated to one of three arms for the study, consisting of the control, personal benefit, and social benefit conditions as described above. Each study arm received two identical messages, one 6 days before data collection started for round 4, and one a single day before data collection started for round 4, which took place from June 13–17, 2020. The study design is shown in **Fig 1.** The previous three rounds of data collection took place in early April, mid-April, and mid-May of 2020. All participants received an ID number to maintain confidentiality and not be identified. Steps were taken to minimize bias by randomly assigning messages and following up soon after the messages were sent to ensure minimal recall bias. The SMS messages were sent in English and Swahili to ensure they were accessible, and generally literacy rates in Nairobi are quite high. The same individual was surveyed each time based on the ID number. Please find supporting information including the STROBE research checklist (S1 Checklist), PLOS One Clinical studies Checklist (S2 Checklist), and the inclusivity questionnaire (S1 Questionnaire).

## Study outcomes

Our primary outcomes were the knowledge of information transmitted in the text messages (recall), stated attitudes towards performing home isolation and caregiving behaviors, and behavioral intentions to isolate and care for those suspected or confirmed to be infected. Secondary outcomes were perceptions of how those in the community would treat or stigmatize

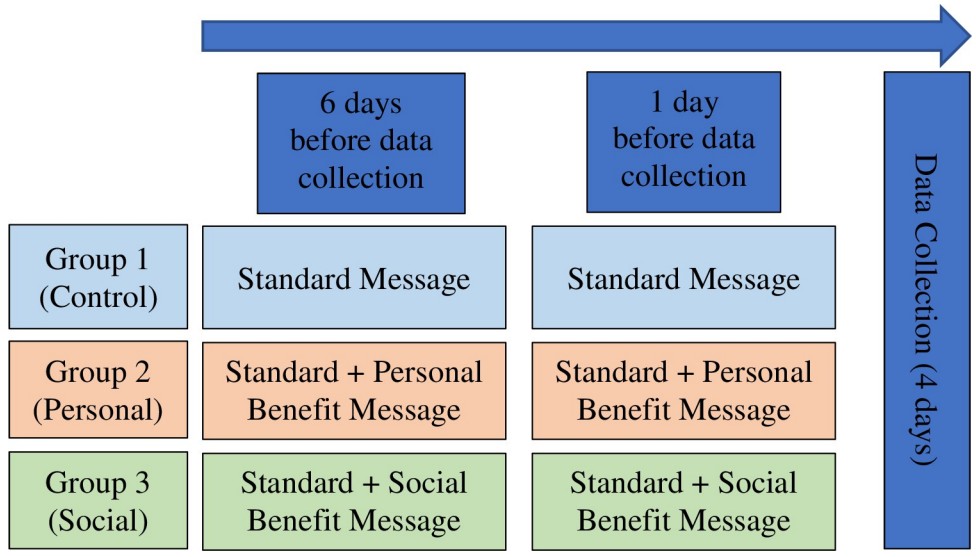

**Fig 1. Study design.**

those infected or suspected of being infected. These were assessed through responses to our survey questions collected via phone calls when the survey was conducted.

## Data analysis

We calculated standard descriptive statistics for demographic factors, knowledge and perceived risk, and attitudes towards COVID-19 and caregiving for all participants. We also analyzed the reach of the text message campaign by calculating the mean number of messages on COVID-19 received from all sources, breaking down the proportion who received messages from each of those sources, and assessing recall of a message on home-based caregiving. We also assessed differences in recall of our messages by stated number of text messages received related to COVID-19 in the prior week to understand how recall and potential impact were affected by message inundation. We further assessed changes in attitudes towards stigma between rounds 3 and 4 using two-proportion, two-tailed z-tests, capturing primarily changes taking place in the population due to considerations beyond our intervention. Finally, we assessed differences between study arms in recall of specific aspects of messages and attitudes related to personal motivations and perceptions of others.

We then conducted an adjusted multiple linear regression analysis of our primary outcome, using both an intention-to-treat (ITT) and secondarily, a per-protocol (PP) approach. Adjusted analyses were conducted to assess any imbalance within the sample due to attrition, as well as to assess the potential association of covariates with message effectiveness. We included several individual demographic characteristics as well as measures of perceived risk or severity of infection, the presence of someone currently or previously infected within or outside the household known personally by the respondent. The per-protocol analysis was conducted to attempt to separate the issue of reach (how many participants received text messages) with the impact of messages received, though this should be taken as exploratory, both because of the nature of per protocol analysis, and because message receipt was self-reported and may be conflated with low recall of the messages.

## Ethical considerations

All participants had been enrolled in person and given informed consent to participate in the AGI-K and NISITU studies and to be re-contacted in the future. Participants reached by phone were informed that this study was related to COVID-19 knowledge, attitudes, and practices; that their participation was voluntary and could be stopped at any time; and that their participation in the other studies was not contingent on being enrolled in the current COVID-19 study. Participants gave informed consent to participate in the COVID-19 study. This study received ethical approval from both the Population Council IRB (p936) and AMREF ESRC (P803/2020) (S1 and S2 Checklists).

## Results

Of the 1910 participants we attempted to contact from the cohort, a total of 1525 people responded to our survey. Most of this gap was due to challenges reaching the person (n = 314, 16.3%), with only 67 refusing to participate (3.5%). Baseline demographic characteristics were generally well balanced across groups, though those in the control group were slightly younger and those in the personal benefit group slightly older (Table 1). Overall, 47.7% of respondents

**Table 1. Basic demographic and psychographic characteristics by study arm (n = 1525).**

| Variable | Control Group | Personal Benefit Group | Social Benefit Group | p-value |
|---|---|---|---|---|
| N | 494 | 515 | 516 | |
| Age (mean (SD)) | 35.13 (10.94) | 37.63 (11.56) | 36.59 (11.30) | 0.002 |
| Sex = Female (%) | 62.9% | 61.1% | 61.0% | 0.793 |
| Education Group | | | | 0.296 |
| No school | 3.4% | 3.3% | 3.1% | |
| Primary | 36.8% | 43.7% | 38.4% | |
| Secondary | 44.3% | 41.7% | 44.4% | |
| Higher | 15.4% | 11.3% | 14.1% | |
| Wealth Quintile | | | | 0.155 |
| 1 | 18.6% | 22.9% | 17.6% | |
| 2 | 22.7% | 16.3% | 20.9% | |
| 3 | 19.4% | 22.5% | 20.2% | |
| 4 | 32.0% | 31.7% | 33.1% | |
| 5 | 7.3% | 6.6% | 8.1% | |
| Lost job/income due to COVID-19 | | | | 0.767 |
| No | 13.8% | 11.5% | 14.0% | |
| Yes, Partial | 38.5% | 40.2% | 39.1% | |
| Yes, Full | 47.8% | 48.3% | 46.9% | |
| HH member 58 or older in home | 16.6% | 13.6% | 12.6% | 0.168 |
| Perceived chance of being infected | | | | 0.686 |
| High | 39.0% | 38.3% | 43.4% | |
| Medium | 26.3% | 27.1% | 25.7% | |
| Low | 27.7% | 26.7% | 24.2% | |
| No Risk | 7.0% | 7.9% | 6.7% | |
| Believes elderly are at risk | 45.7% | 49.8% | 46.8% | 0.433 |
| Knows someone with COVID-19 | | | | 0.475 |
| Yes | 5.9% | 4.3% | 4.8% | |
| Knows someone suspected (not yet tested) | 0.8% | 1.7% | 1.0% | |
| No | 93.3% | 94.0% | 94.2% | |
| Has a place to care for someone infected in the home | 7.7% | 12.0% | 10.7% | 0.066 |

had experienced a full loss of job and/or income and 39.3% had experienced only a partial job loss. Almost half (47.5%) knew that the elderly were at elevated risk of more severe COVID-19, and 14.2% had at least one family member aged 58 or older living in the household. Only 5% knew someone who had tested positive for COVID-19, and 1.2% knew someone suspected of being infected, but not yet tested. Only about 10% of people reported having a suitable place to care for someone with COVID-19 in the home.

Respondents reported significant concern regarding social exclusion and stigma prior to and during the study period (Table 2). In the pre-intervention round of data collection in May 2020, a strong majority of participants reported that if they were infected, people would stop talking to them (82%) and stop visiting their house or business (92%). Though these might be reasonable precautions in some situations, similarly high proportions reported that people would gossip about them (92%) and would treat their family badly (73%). Less than half reported that others would bring them the food (42%) or medicines (37%) they needed. These rates were relatively stable between rounds (one month apart; May to June 2020), where any differences might have been due to changes in the context or due to the messaging, though people were more likely to report that others would bring them necessary food (51% vs 42%, p < .001) or medicine (46% vs 37%, p < .001). While not asked during round 3 (May 2020), many in the round 4 survey (June 2020) reported that people in the community would avoid them if they believed that if they were to be infected in the future, after the COVID-19 infection cleared, people in the community would still avoid them (53%), that they could not return to work (35%), and that their family would not welcome them back into their house (26%).

Earlier rounds of the survey had revealed that more than 90% of the cohort had received text messages related to COVID-19 prior to our intervention. Respondents reported receiving more than four messages in the past week related to COVID-19, with their mobile service provider, the government/Ministry of Health, and NGOs being the most common sources outside of the study messages. About 60% reported receiving a message from the study, identified as either coming from Population Council and/or MPayer, with no differences between study arms (Table 3). However, an even higher number reported receiving a message about caring for those infected when directly prompted, with significantly higher rates of recall for those in the private (80.0%) or social (77.2%) benefit condition than those in the control condition (70.8%, p = .005). As 23.8% of respondents did not recall receiving a message with the content of the intervention messages, an additional exploratory analysis of the impact of messages on those recalling receiving them was also conducted to understand whether challenges with

**Table 2. Behavioral expectations if the respondent hypothetically had been known to contract COVID-19 (includes all respondents in round 3 and round 4 of the survey).**

|  | Round 3 | Round 4 | p-value |
|---|---|---|---|
| n = | 1748 | 1525 |  |
| People would stop talking to me | 82.3% | 80.8% | 0.297 |
| People would stop visiting my house/business | 91.6% | 89.1% | 0.016 |
| People would gossip about me | 91.5% | 92.1% | 0.625 |
| People in the community would treat my family badly | 73.1% | 69.2% | 0.016 |
| People I know would bring me food I need | 41.8% | 50.6% | <0.001 |
| People I know would bring me the medicines I need | 36.6% | 45.8% | <0.001 |
| After I have recovered from Corona virus, people in the community would still avoid me | NA | 53.0% | NA |
| After I no longer had coronavirus, my employer would not take me back to work | NA | 34.8% | NA |
| I would not be welcome back into my house by family | NA | 26.2% | NA |

**Table 3. Reach of text messages in target population (n = 1,525).**

| Variable | Control Group | Private Benefit Group | Social Benefit Group | p-value |
|---|---|---|---|---|
| N = | 494 | 515 | 516 | |
| Number of messages received in past week (SD) | 4.30 (7.63) | 4.62 (7.34) | 4.26 (5.44) | 0.658 |
| Recalled sources of messages received in past week | | | | |
| Population Council/Mpayer* | 58.3% | 62.9% | 61.0% | 0.321 |
| Government/Ministry of Health | 38.5% | 37.7% | 39.5% | 0.862 |
| Family/Friends | 3.1% | 3.3% | 3.5% | 0.947 |
| NGO | 19.1% | 18.9% | 19.1% | 0.994 |
| Church/mosque | 1.2% | 0.9% | 1.8% | 0.491 |
| Unknown source | 6.0% | 4.4% | 5.7% | 0.528 |
| Mobile Service Provider | 42.3% | 37.9% | 44.1% | 0.153 |
| Other | 1.7% | 2.0% | 2.6% | 0.598 |
| Remembers message about caring for the infected | 70.8% | 80.0% | 77.2% | 0.005 |

Notes: *Source of intervention messages

reach may have impacted the observed results. Table 3 includes all 1,525 individuals in round 4, conducted in June 2020.

In this table, we report responses from the 1,160 individuals who participated in the randomized evaluation. While 1,525 received messages, we aimed to conduct a follow up survey with a subset. Respondents were asked to report what they recalled from the text message received (Table 4). All three conditions received information that (1) anyone could be infected, (2) people should self-isolate, and (3) those infected should still be loved and cared for. While there were no differences between arms for the first two aspects of the message, recall of the third aspect was higher in the private benefit condition (51.1%) than in the control (39.1%, p < .001). Recall of the private benefit condition's additional message was also higher in that group than in the control (21.9% vs 13.2%, p < .001), while recall of the additional message from the social benefit condition was not higher than the control. However, there was no difference between study arms for most attitudes or for behavioral intentions, with about two-thirds of respondents across arms reporting that they would be willing to care for an infected family member.

We assessed the associations of covariates with the primary outcome through an intention to treat (ITT) and per-protocol (PP) analysis to understand which characteristics of individuals might be associated with the intention to care for an infected household member (Table 5). There were no significant impacts of the text messages on behavioral intentions for either condition in either the intention to treat or per protocol analysis. Of all covariates, only wealth and having a place to care for an infected person were significantly associated with the primary outcome. Those with a place to care for an infected person were 18.8 percentage points more likely to express an intention to care for an infected person at home, while those in wealth quintiles 3–5 were about 15 percentage points less likely to report an intention to care for a family member who was infected compared to those in the lowest wealth quintile.

## Discussion

This study was conducted early during a global pandemic, when there was fear and the potential for stigma associated with infection to lead to behaviors harmful to those infected and those around them. Despite this, we found that intentions to care for infected household members at home was quite high, though beliefs that others would stigmatize or otherwise act

**Table 4.** Differences in knowledge, attitudes, and behavioral intentions by study arm, among those included in the RCT (n = 1,160).

| Variable | Control Group | Private Benefit Group | p-value | Social Benefit Group | p-value |
|---|---|---|---|---|---|
| **Recall (of those who remembered receiving the intervention message)** | **n = 350** | **n = 412** | | **n = 398** | |
| Anyone can get infected with Coronavirus | 14.2% | 16.9% | 0.235 | 15.9% | 0.452 |
| Those infected should self-isolate as much as possible | 8.7% | 8.5% | 0.930 | 10.5% | 0.335 |
| Those infected should still be loved, cared for and accepted by friends and neighbors | 39.1% | 51.1% | < .001 | 45.2% | 0.051 |
| You should treat others with Coronavirus how you would like to be treated | 13.2% | 21.9% | < .001 | 15.1% | 0.403 |
| Supporting one another will help our community and nation through this difficult time | 8.1% | 9.3% | 0.499 | 9.7% | 0.378 |
| **Attitudes and Motivations** | | | | | |
| Main factor influencing decisions regarding COVID-19 | | | | | |
| Economic/need to make money | 35.6% | 31.8% | 0.200 | 30.6% | 0.090 |
| Keep myself health/safe from getting Corona | 10.7% | 15.0% | 0.049 | 14.5% | 0.076 |
| Keep my family safe from getting Corona | 22.9% | 20.0% | 0.266 | 21.3% | 0.547 |
| Keep the community/country safe from getting corona | 4.9% | 6.2% | 0.359 | 6.4% | 0.298 |
| A desire to have life go back to normal | 25.9% | 26.8% | 0.750 | 26.7% | 0.764 |
| Other | 0.0% | 0.2% | 0.487 | 0.4% | 0.165 |
| Perceptions of Others' Response | | | | | |
| People would stop talking to me | 80.8% | 81.2% | 0.873 | 80.4% | 0.890 |
| People would stop visiting my house/business | 90.1% | 88.7% | 0.494 | 88.6% | 0.440 |
| People would gossip about me | 93.3% | 90.9% | 0.151 | 92.1% | 0.457 |
| People in the community would treat my family badly | 68.4% | 69.9% | 0.611 | 69.4% | 0.742 |
| People I know would bring me food I need | 51.0% | 50.1% | 0.772 | 50.8% | 0.940 |
| People I know would bring me the medicines I need | 44.7% | 46.8% | 0.512 | 45.9% | 0.704 |
| After I have recovered from Corona virus, people in the community would still avoid me | 68.2% | 66.6% | 0.589 | 63.4% | 0.104 |
| After I no longer had coronavirus, my employer would not take me back to work | 43.1% | 45.2% | 0.496 | 41.5% | 0.598 |
| I would not be welcome back into my house by family | 24.7% | 24.9% | 0.954 | 28.9% | 0.131 |
| **Behavioral Intentions** | | | | | |
| Would be willing to care for infected household member | 70.0% | 69.3% | .806 | 64.5% | .061 |

harmfully towards the households of those infected were widespread. We detected some impacts of different messaging conditions, with increased knowledge of transmitted information for the private benefit/reciprocity condition. This actually increased the more messages the person recalled receiving, in contrast to the social benefit or information only conditions, where knowledge decreased as more messages were received. Exploratory analysis revealed that the most significant factors associated with intention to care for an infected household member at home were related to capabilities (wealth and sufficient space to care for an infected person), rather than motivations or opportunities/reminders.

While it is reasonable that governments may want to widely disseminate information and even persuasive messaging through text messages as an inexpensive and rapid form of informing the public during an emergency, our results suggest that this may have only limited effects depending on the type of behavior. While previous studies have shown some behaviors may be increased through improving motivation or opportunity [22], those driven largely by contextual factors may not be amenable to such interventions. Even though motivational messages targeting a reciprocity motive were more likely to be recalled compared to other conditions and were less likely to be crowded out by the large volume of informational messages being broadcast, they had little effect and other interventions were certainly needed; on the other

**Table 5. Primary outcome results with covariates: Intention to treat and per protocol analyses (n = 1,160).**

| Variable | ITT Analysis | | PP Analysis | |
|---|---|---|---|---|
| | Parameter Estimate | p-value | Parameter Estimate | p-value |
| (Intercept) | 0.820 | <0.001 | 0.815 | <0.001 |
| Behavioral Intention: Personal Benefit Group | -0.013 | 0.665 | -0.048 | 0.337 |
| Behavioral Intention: Social Benefit Group | -0.042 | 0.171 | -0.019 | 0.703 |
| Age (mean (SD)) | 0.000 | 0.856 | 0.000 | 0.939 |
| Sex = Female (%) | -0.001 | 0.959 | 0.001 | 0.964 |
| Education: Primary | -0.038 | 0.613 | -0.031 | 0.683 |
| Education: Secondary | 0.010 | 0.900 | 0.019 | 0.801 |
| Education: Higher Ed | 0.022 | 0.794 | 0.031 | 0.709 |
| Wealth Quintile: 1 | ref | ref | ref | r |
| Wealth Quintile: 2 | -0.073 | 0.066 | -0.072 | 0.071 |
| Wealth Quintile: 3 | -0.146 | <0.001 | -0.142 | <0.001 |
| Wealth Quintile: 4 | -0.158 | <0.001 | -0.156 | <0.001 |
| Wealth Quintile: 5 | -0.159 | 0.004 | -0.154 | 0.005 |
| Job Loss: None | ref | ref | ref | ref |
| Job Loss: Partial | -0.070 | 0.085 | -0.068 | 0.096 |
| Job Loss: Full | -0.050 | 0.214 | -0.047 | 0.242 |
| HH member 58 or older in home | -0.008 | 0.817 | -0.009 | 0.790 |
| Perceived chance of being infected: Medium | -0.022 | 0.474 | -0.021 | 0.492 |
| Perceived chance of being infected: Low | 0.015 | 0.631 | 0.014 | 0.664 |
| Perceived chance of being infected: None | 0.081 | 0.114 | 0.078 | 0.125 |
| Believes elderly are at risk | 0.004 | 0.884 | 0.006 | 0.807 |
| Knows someone with COVID-19: Yes | -0.035 | 0.542 | -0.039 | 0.500 |
| Knows someone with COVID-19: Suspected | 0.046 | 0.686 | 0.040 | 0.723 |
| Has a place at home to care for infected person | 0.188 | <0.001 | 0.189 | <0.001 |
| Thinks family will be stigmatized for caring for infected person | 0.043 | 0.143 | 0.043 | 0.140 |
| Received Message | | | -0.024 | 0.592 |
| Received Message * Personal Benefit Group | | | 0.056 | 0.380 |
| Received Message * Social Benefit Group | | | -0.037 | 0.552 |

hand, more intensive motivational campaigns using multimedia may have been more effective [11]. Our study aims to gauge behavioral intent, with potential implications for behaviors, but we were limited in data collection during the pandemic. Overall, messaging alone may not have been sufficient to shift behaviors particularly during a very uncertain time and when COVID-19 transmission and risks were not well understood.

Another limitation of a text message campaign is that there is little sense of "common knowledge" needed to drive social norms change. It is striking that there were no differences observed in nine different beliefs about others' perceptions for either condition, especially for the social benefit condition. While text messages may be a useful vehicle for rapid information dissemination, they may be less effective in creating empirical or normative expectations required to affect social norms [23]. Relatedly, our respondents reported receiving many different messages from different sources. Early in the pandemic when little was understood about COVID-19, people may have been overwhelmed by different (often conflicting) information and high frequency of messages, leading to confusion and overload of information [24].

While nudges in a variety of domains have been found to be cost-effective [25], there have been numerous calls to understand which kinds of nudges work best in which circumstances

or for which behaviors [26]. Unfortunately, little predictive theory or understanding of behavioral kinds [27] has been applied to understand when or if nudges will be effective. This study suggests that those deploying behavioral interventions need to more carefully consider whether nudges or other kinds of interventions will be effective for a desired outcome. In this case as well, messages and intentions may not have been sufficient to overcome challenges related to misinformation early in the pandemic, and the combination with structural barriers to uptake of the behavior. In slums, housing is often overcrowded, with poor water and sanitation and low ventilation [28]. Inadequate housing infrastructure for the urban poor made it difficult if not impossible to quarantine someone within the home and likely contributed to the rapid spread of COVID-19. Structural barriers also related to economic ones, where the most economically vulnerable faced evictions, food insecurity, and could not quarantine due to the loss of income [28, 29]. Even with intentions to care for a sick family member, structural and economic constraints may have been significant barriers.

There are several limitations to note in this study. First, there was not a "pure" control condition, as the government had already decided to send messages, and thus the only valid question to consider was the differential impacts of various kinds of messages. We were unable to compare to a group that had received no text message since everyone in Nairobi was receiving them. However, this study aimed to measure an added effect using social or personal benefit messages, not necessarily the impact of receiving any message. Second, there were slightly fewer respondents than anticipated by our original sample size calculation, but the overall findings of our study do not appear to be significantly affected. Third, the intervention was quite light touch, with only two brief text messages being sent, and so little can be concluded about what a more intensive campaign might have done in this setting—however, such campaigns would be difficult for governments in LMIC settings to deliver. Relatedly, we rely on self-reported responses to the questions and could only measure behavioral intention not actual behaviors. Lastly, we leveraged existing cohorts of households in Nairobi, but this means our sample is inherently not representative. We included households with at least one adolescent household member, so households with only one person, or those with only elderly household members, or only very young children, were not included.

## Conclusion

Text messages are inexpensive and may rapidly and easily reach large numbers of people in emergency settings. Furthermore, leveraging behavioral theory to design the persuasive or motivational messages used may increase information recall. However, for addressing more complex attitudes emerging in unfamiliar situations or when contextual factors constrain behaviors, such messages may have little ability to drive significant behavior change. Thus, more comprehensive interventions targeting motivations or capabilities (such as alternatives to home care in crowded, low-income communities) need to be considered by policymakers.

## Supporting information

**S1 Checklist. STROBE research checklist.**
(DOCX)

**S2 Checklist. PLOS One clinical studies checklist.**
(DOCX)

**S1 Questionnaire. Inclusivity questionnaire.**
(DOCX)

## Acknowledgments

The authors would like to acknowledge the data collection team based in Nairobi and the participants for their time.

## Author Contributions

**Conceptualization:** James B. Tidwell, Jessie Pinchoff, Timothy Abuya, Karen Austrian.

**Data curation:** Eva Muluve, Daniel Mwanga, Faith Mbushi.

**Formal analysis:** James B. Tidwell, Daniel Mwanga.

**Funding acquisition:** Karen Austrian.

**Investigation:** James B. Tidwell, Timothy Abuya, Eva Muluve, Daniel Mwanga, Karen Austrian.

**Methodology:** Jessie Pinchoff, Daniel Mwanga, Faith Mbushi.

**Project administration:** Eva Muluve, Daniel Mwanga, Faith Mbushi.

**Software:** Eva Muluve.

**Supervision:** Faith Mbushi, Karen Austrian.

**Writing – original draft:** James B. Tidwell.

**Writing – review & editing:** Jessie Pinchoff, Timothy Abuya, Eva Muluve, Daniel Mwanga, Faith Mbushi, Karen Austrian.

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
