## [Decision Letter · Decision Letter 0]

16 Aug 2023

PONE-D-23-11319The limits of nudging: Results of a randomized trial of text messages to promote home-based caregiving and reduce perceptions of stigma for COVID-19 patients in Kenyan informal settlementsPLOS ONE

Dear Dr. Pinchoff,

Thank you for submitting your manuscript to PLOS ONE. After careful consideration, we feel that it has merit but does not fully meet PLOS ONE’s publication criteria as it currently stands. Therefore, we invite you to submit a revised version of the manuscript that addresses the points raised during the review process.

As the academic editor, I would first like to apologise for the delay in responding. It was very challenging to identify additional reviewers to review the manuscript. Based on the feedback from reviewer 1, I suggest that you address these major comments and if interested re-submit a revised version. 

We look forward to receiving your revised manuscript.

Kind regards,

Candice Maylene Chetty-Makkan, MA, PhD

Academic Editor

PLOS ONE

Journal Requirements:

Reviewers' comments:

Reviewer's Responses to Questions

**Comments to the Author**

1. Is the manuscript technically sound, and do the data support the conclusions?

Reviewer #1: No

2. Has the statistical analysis been performed appropriately and rigorously? 

Reviewer #1: I Don't Know

3. Have the authors made all data underlying the findings in their manuscript fully available?

Reviewer #1: Yes

4. Is the manuscript presented in an intelligible fashion and written in standard English?

Reviewer #1: No

5. Review Comments to the Author

Reviewer #1: This manuscript reports on a text message intervention implemented in Kenya. The study tested the effect of text messaging on message recall, attitudes, intentions, and stigma related to COVID-19.

Include the hypothesis in the manuscript abstract to clarify the dependent variable(s).

Background

Paragraph 1

- Review sentence starting with, “Initially, all…”

- Ideally, the measures employed/ implemented (by Kenyan MoH) should be referenced.

Paragraph 2

- It may be worth clarifying what the significant toll was.

- Clarify the last sentence; did C19 messaging cause community to socially isolate people with SC2, and also stigmatize healthcare workers?

Paragraph 3

- The effect of text messaging to influence health behaviour is variable and it may be instructive for the reader to understand this going into the methods section. There are a number of systematic reviews on text messages and health behavior that may be worth covering in the introduction.

Additionally, provide evidence for the hypotheses that will be tested – ie the effect of the independent variable(s) (text message content) upon the dependent variables 1) recall, 2) attitudes, 3) intentions 4) stigma.

Methods

- Data analysis – state what the measures were, and how the outcome was measured. How were knowledge, attitudes, and intentions measured? Was there a hypothesis? What was the hypothesis? The hypothesis seems inferred; that texts containing personal and social motivations will increase knowledge, change attitudes, and increase intentions to care for household contacts.

- It may be worth clarifying the hypothesis/ hypotheses, reflecting back on the evidence/ literature provided in the introduction.

Discussion

- The “no control condition” is a major limitation on the study. It may be worth explaining the impact on the outcome variables. What is the extent of the influence of non-study text messages, and other information on the dependent variables, recall, attitudes, intentions, stigma? To what extent is it possible, considering this limitation, to draw conclusions and make inferences about the intervention (independent variable)?

There are a few typographical/ sentence structure errors that can easily be addressed.

6. PLOS authors have the option to publish the peer review history of their article (what does this mean?). If published, this will include your full peer review and any attached files.

Reviewer #1: **Yes: **Philip Smith

---

## [Author Response · Author response to Decision Letter 0]

12 Sep 2023

Submitted responses as an attachment, but thank you again for the opportunity to revise and strengthen our paper.

Thank you!

---

## [Decision Letter · Decision Letter 1]

6 Feb 2024

PONE-D-23-11319R1

The limits of nudging: Results of a randomized trial of text messages to promote home-based caregiving and reduce perceptions of stigma for COVID-19 patients in Kenyan informal settlements

PLOS ONE

Dear Dr. Pinchoff,

Thank you for submitting your manuscript to PLOS ONE. After careful consideration, we feel that it has merit but does not fully meet PLOS ONE’s publication criteria as it currently stands. Therefore, we invite you to submit a revised version of the manuscript that addresses the points raised during the review process.

**ACADEMIC EDITOR: **Thank you for your patience as we struggled to find reviewers over the holiday period.

This article has merit, but requires additional content and restructuring to ensure that readers can follow the story and also that there is enough detail to enable replication or adaptation with future designs. With regards to adding an additional table by one review, I do not require this, though you may want to accept this suggestion. 

Some specifics I will be looking for in the revised manuscript: 

As noted by both reviewers, ensure that the title aligns with the content, in particular the term 'nudge.' Preferably add further detail about your definition(s) of nudging in the body of the manuscript.To address reviewer questions about the theoretical basis for selection of private and social benefit frames, at a minimum include references in the sentence that begins 'Based on existing literature' on p.4 [NB: references 17-20 that were cited on p.5 seem relevant here]Clarify what is meant by informed in the sentence 'sent informed participants' on p.4. Consider rewordingWhile I assume English, specify the language of messages in the behavioral intervention section (p.4-5) and address literacy rates of the sample (if known) in the section on sample (p.5)Consider reframing the title for Table 3. The statements present behavioral expectations vs attitudes towards stigma (p.9)Ensure the basis for the denominators for Tables 4-6 are clear in the text, to ensure accurate interpretation. For instance, are the numbers lower in Table 4 because only those who recalled messages from the Population Council/Mpayer (see Table 2) were included? Specify which messages in Tables 4-6 belong to control/all vs the private or social benefit arms. This can be done next to the statements in the first column.Given the study design (Figure 1), where private and social benefit messages were only sent to one arm, how does the research team explain recall of the same messages among both control and other arms as presented in Tables 4-5? This needs to be unpacked more in discussion. Similarly the fact that only 2 messages were sent to each group but they recalled more messages needs to be addressed overtly in the discussion.Further to a reviewers request for more clarity on Table 5, move the current column 1 heading above the other 4 columns and create new headings for both the table and first column that reflects contentIn the discussion, paragraph 2, p.13, ensure that your study results are discussed in the context of gauging behavioral intention and not actual behavior. The discussion should also expand on findings about structural determinants (wealth and space) on intention to care (Table 6 results) as part of the 'other kinds of interventions' mentioned at the bottom of p.13

We look forward to receiving your revised manuscript.

Kind regards,

Sara Jewett Nieuwoudt, Ph.D, MPH

Academic Editor

PLOS ONE

Reviewers' comments:

Reviewer's Responses to Questions

**Comments to the Author**

1. If the authors have adequately addressed your comments raised in a previous round of review and you feel that this manuscript is now acceptable for publication, you may indicate that here to bypass the “Comments to the Author” section, enter your conflict of interest statement in the “Confidential to Editor” section, and submit your "Accept" recommendation.

Reviewer #2: (No Response)

Reviewer #3: (No Response)

2. Is the manuscript technically sound, and do the data support the conclusions?

Reviewer #2: Yes

Reviewer #3: Partly

3. Has the statistical analysis been performed appropriately and rigorously? 

Reviewer #2: I Don't Know

Reviewer #3: Yes

4. Have the authors made all data underlying the findings in their manuscript fully available?

Reviewer #2: Yes

Reviewer #3: Yes

5. Is the manuscript presented in an intelligible fashion and written in standard English?

Reviewer #2: Yes

Reviewer #3: Yes

6. Review Comments to the Author

Reviewer #2: This research makes an important contribution towards understanding the limitations of using SMS to drive behaviour change. However, the paper could be improved in some areas to make it clearer to practitioners how they should go about intervention design through SMS and what the benefits and limitations are. It would be good to get a brief overview of the literature and what we have learnt about nudging. It was also difficult to understand how the team went about designing their own intervention. I have attached comments related to specific sections in the paper, and here is a summary:

- The title is making a large statement about nudging. It would be good to get some background information in the introduction section about nudging. What constitutes a nudge? Under what circumstances is it supposed to work, and under which is it not supposed to from a theoretical perspective? What are the merits and what are the limitations?

- It would be good to have a clearer understanding of the theoretical basis that the team used to inform the design of their SMS and why specific behavioural determinants were focused on.

- In the discussion it would be good to get some insights on whether the messenger effect is at play: so would the same messages be more effective if they came from another source, like a WhatsApp group?

- Consider adding some information on alternative behaviour change techniques that could be used through other modalities that may be more effective from a theoretical perspective.

- How would the others suggest that governments go about public health mass media communication in an effective way? What has worked better in other countries?

Here are the comments per page:

Page 10: Could you give a clearer theoretical background to the intervention design? Could you be a bit clearer on why these behavioural determinants were chosen? What theories informed your intervention? Were different phrasings pre-tested before this was chosen? Were the messages translated?

Page 11: Can you add a table summarizing some of the information that led to the SMS phrasing decisions? Please explain how the surveys administered. Were the survey questions for attitudes and intentions based on existing scales or guidance? If so, which ones?

Page 17: The information on Table 5 is not clear, could you present it differently?

Reviewer #3: Thank you for the opportunity to review this paper. Exploring the use of tailored messaging to improve outcomes in a cost-effective way is a pertinent topic and the title is good.

The article could benefit from a better structured and written background. Please check grammar and write for clarity e.g. adding years to dates. It should also follow a more coherent “story” or focus e.g. text messages are a cheap intervention which has potential for large reach but unknown if tailored messaging is better than plain informational messaging... Please link to COM-B model if use this in discussion. The hypothesis is clear.

More information is needed on the measures in the methods section, especially in describing outcomes and how recall of the PC-specific (vs government and other) messages was measured. It is unclear who, in the households, was interviewed - the phone owner? There were IDs but how was this ensured that it was the same person? What are the implications of using pre-identified HH with adolescents vs general community? What about exposure to other messaging beyond text e.g. mass media? Was this not measured? The last sentence of the existing measures section should probably be in limitations.

In the discussion, please link back to the background more (with a clearer "story") and to the framework used. Is the conclusion about not being amenable right? iShould it not be about needing to explore whether messaging needs to be either accompanied by practical information on how to address those contextual factors or a structural intervention such as a grant?

The title is nice but there is limited use of the term "nudge" in ther article itself.

This article has potential but requires some more work.

7. PLOS authors have the option to publish the peer review history of their article (what does this mean?). If published, this will include your full peer review and any attached files.

Reviewer #2: **Yes: **Sarah Osman

Reviewer #3: No

---

## [Author Response · Author response to Decision Letter 1]

9 Apr 2024

we have uploaded our response to reviewers. Thank you for the opportunity to revise the manuscript.

---

## [Decision Letter · Decision Letter 2]

27 May 2024

The limits of nudging: Results of a randomized trial of text messages to promote home-based caregiving and reduce perceptions of stigma for COVID-19 patients in Kenyan informal settlements

PONE-D-23-11319R2

Dear Dr. Pinchoff,

We’re pleased to inform you that your manuscript has been judged scientifically suitable for publication and will be formally accepted for publication once it meets all outstanding technical requirements.

Kind regards,

Sara Jewett Nieuwoudt, Ph.D, MPH

Academic Editor

PLOS ONE

Additional Editor Comments:

Thank you for your resubmission. Given difficulty in getting a second reviewer, I have taken it upon myself to do a final review of your manuscript (as Reviewer #1) to facilitate the publication of your study. Please note the two suggestions from Reviewer #2. The specific reference you add is at your discretion. You are not obligated to use one of the two suggested. The second point on Table numbering needs to be acted upon before going to press. I am satisfied that you have addressed all other comments and that your manuscript adheres to our publication criteria.

Reviewers' comments:

Reviewer's Responses to Questions

**Comments to the Author**

1. If the authors have adequately addressed your comments raised in a previous round of review and you feel that this manuscript is now acceptable for publication, you may indicate that here to bypass the “Comments to the Author” section, enter your conflict of interest statement in the “Confidential to Editor” section, and submit your "Accept" recommendation.

Reviewer #1: All comments have been addressed

Reviewer #2: All comments have been addressed

2. Is the manuscript technically sound, and do the data support the conclusions?

Reviewer #1: Yes

Reviewer #2: Yes

3. Has the statistical analysis been performed appropriately and rigorously? 

Reviewer #1: Yes

Reviewer #2: I Don't Know

4. Have the authors made all data underlying the findings in their manuscript fully available?

Reviewer #1: Yes

Reviewer #2: Yes

5. Is the manuscript presented in an intelligible fashion and written in standard English?

Reviewer #1: Yes

Reviewer #2: Yes

6. Review Comments to the Author

Reviewer #1: Thank you for taking the time to revise and resubmit. I recommend a final round of proofreading, e.g.  in Table 5, 'ref' is not written out for Wealth Quintile: 1 in the final column. However, this is minor and does not require another round of review. 

Reviewer #2: Thank you for your responses and this new draft. My only recommendation is to add a reference where you say 'We employed a behavioral nudging approach...' on page three. Please add a reference to either: Thaler, R. H., & Sunstein, C. R. (2008). Nudge: Improving decisions about health, wealth, and happiness; or to Ridder, D.T. (2014). Nudging for beginners. The European health psychologist, 16, 1-6. This will make it easier for the reader to understand what you mean by nudging, even if the term is widely used.

I believe Table 4 is misnumbered, currently reads as Table 2.

7. PLOS authors have the option to publish the peer review history of their article (what does this mean?). If published, this will include your full peer review and any attached files.

Reviewer #1: No

Reviewer #2: **Yes: **Sarah Osman

---

## [Editor Report · Acceptance letter]

19 Jul 2024

PONE-D-23-11319R2 

PLOS ONE

Dear Dr. Pinchoff, 

I'm pleased to inform you that your manuscript has been deemed suitable for publication in PLOS ONE. Congratulations! Your manuscript is now being handed over to our production team.

Kind regards, 

on behalf of

Dr. Sara Jewett Nieuwoudt 

Academic Editor

PLOS ONE